# Meta Adversarial Perturbations

**Chia-Hung Yuan**[1,2], **Pin-Yu Chen**[1,3], **Chia-Mu Yu**[2]

[1]MIT-IBM Watson AI Lab
[2]National Yang Ming Chiao Tung University
[3]IBM Research
jimmy.chyuan@gmail.com, pin-yu.chen@ibm.com, chiamuyu@nycu.edu.tw

## Abstract

A plethora of attack methods have been proposed to generate adversarial examples, among which the iterative methods have been demonstrated the ability to find a strong attack. However, the computation of an adversarial perturbation for a new data point requires solving a time-consuming optimization problem from scratch. To generate a stronger attack, it normally requires updating a data point with more iterations. In this paper, we show the existence of a *meta adversarial perturbation* (MAP), a better initialization that causes natural images to be misclassified with high probability after being updated through only a one-step gradient ascent update, and propose an algorithm for computing such perturbations. We conduct extensive experiments, and the empirical results demonstrate that state-of-the-art deep neural networks are vulnerable to meta perturbations. We further show that these perturbations are not only image-agnostic, but also model-agnostic, as a single perturbation generalizes well across unseen data points and different neural network architectures.

## 1 Introduction

Deep neural networks (DNNs) have achieved remarkable performance in many applications, including computer vision, natural language processing, speech, and robotics, etc. However, DNNs are shown to be vulnerable to adversarial examples (Szegedy et al. 2013; Goodfellow, Shlens, and Szegedy 2014), i.e. examples that are intentionally designed to be misclassified by the models but nearly imperceptible to human eyes. In recent years, many methods have been proposed to craft such malicious examples (Szegedy et al. 2013; Goodfellow, Shlens, and Szegedy 2014; Moosavi-Dezfooli, Fawzi, and Frossard 2016; Kurakin et al. 2016; Madry et al. 2017; Carlini and Wagner 2017; Chen et al. 2018), among which the iterative methods, such as PGD (Madry et al. 2017), BIM (Kurakin et al. 2016), and MIM (Dong et al. 2018), have been demonstrated to be effective to craft adversarial attacks with a high success rate. Nevertheless, to craft a stronger attack with iterative methods, it usually requires updating a data point through more gradient ascent steps. This time-consuming process gives rise to a question: is it possible to find a *single* perturbation, which can be served

as a good meta initialization, such that after a few updates, it can become an effective attack for different data points?

Inspired by the philosophy of meta-learning (Schmidhuber 1987; Bengio, Bengio, and Cloutier 1990; Andrychowicz et al. 2016; Li and Malik 2016; Finn, Abbeel, and Levine 2017), we show the existence of a quasi-imperceptible *meta adversarial perturbation* (MAP) that leads natural images to be misclassified with high probability after **being updated through only one-step gradient ascent update**. In meta-learning, the goal of the trained model is to quickly adapt to a new task with a small amount of data. On the contrary, the goal of the meta perturbations is to rapidly adapt to a new data point within a few iterations. The key idea underlying our method is to train an initial perturbation such that it has maximal performance on new data after the perturbation has been updated through one or a few gradient steps. We then propose a simple algorithm, which is plug-and-play and is compatible with any gradient-based iterative adversarial attack method, for seeking such perturbations. By adding a meta perturbation at initialization, we can craft a more effective adversarial example without multi-step updates.

We summarize our main contributions as follows:

- We show the existence of image-agnostic learnable meta adversarial perturbations for efficient robustness evaluation of state-of-the-art deep neural networks.

- We propose an algorithm (MAP) to find meta perturbations, such that a small number of gradient ascent updates will suffice to be a strong attack on a new data point.

- We show that our meta perturbations have remarkable generalizability, as a perturbation computed from a small number of training data is able to adapt and fool the unseen data with high probability.

- We demonstrate that meta perturbations are not only image-agnostic, but also model-agnostic. Such perturbations generalize well across a wide range of deep neural networks.

## 2 Related Works

There is a large body of works on adversarial attacks. Please refer to (Chakraborty et al. 2018; Akhtar and Mian 2018; Biggio and Roli 2018) for comprehensive surveys. Here, we discuss the works most closely related to ours.

## 2.1 Data-dependent Adversarial Perturbations

Despite the impressive performance of deep neural networks on many domains, these classifiers are shown to be vulnerable to adversarial perturbations (Szegedy et al. 2013; Goodfellow, Shlens, and Szegedy 2014). Generating an adversarial example requires solving an optimization problem (Moosavi-Dezfooli, Fawzi, and Frossard 2016; Carlini and Wagner 2017) or through multiple steps of gradient ascent (Goodfellow, Shlens, and Szegedy 2014; Kurakin et al. 2016; Madry et al. 2017; Chen et al. 2018) for each data point independently, among which the iterative methods have been shown to be able to craft an attack with a high success rate. Given a data point $x$, a corresponding label $y$, and a classifier $f$ parametrized by $\theta$. Let $L$ denote the loss function for the classification task, which is usually the cross-entropy loss. FGSM (Goodfellow, Shlens, and Szegedy 2014) utilizes gradient information to compute the adversarial perturbation in one step that maximizes the loss:

$$x' = x + \epsilon \, \text{sign}(\nabla_x L(f_\theta, x, y)), \qquad (1)$$

where $x'$ is the adversarial example and $\epsilon$ is the maximum allowable perturbation measured by $l_\infty$ distance. This simple one-step method is extended by several follow-up works (Kurakin et al. 2016; Madry et al. 2017; Dong et al. 2018; Xie et al. 2019), which propose iterative methods to improve the success rate of the adversarial attack. More specifically, those methods generate adversarial examples through multistep updates, which can be described as:

$$x^{t+1} = \Pi_\epsilon \big( x^t + \gamma \, \text{sign}(\nabla_x L(f_\theta, x, y)) \big), \qquad (2)$$

where $\Pi_\epsilon$ projects the updated perturbations onto the feasible set if they exceed the maximum allowable amount indicated by $\epsilon$. $x^0 = x$ and $\gamma = \epsilon/T$, where $T$ is the number of iterations. To generate a malicious example that has a high probability to be misclassified by the model, the perturbation sample needs to be updated with more iterations. The computational time has a linear relationship with the number of iterations, thus it takes more time to craft a strong attack.

## 2.2 Universal Adversarial Perturbations

Instead of solving a data-dependent optimization problem to craft adversarial examples, (Moosavi-Dezfooli et al. 2017) shows the existence of a universal adversarial perturbation (UAP). Such a perturbation is image-agnostic and quasi-imperceptible, as a single perturbation can fool the classifier $f$ on most data points sampled from a distribution over data distribution $\mu$. That is, they seek a perturbation $v$ such that

$$f(x + v) \neq f(x) \text{ for "most" } x \sim \mu. \qquad (3)$$

In other words, the perturbation process for a new data point involves merely the addition of precomputed UAP to it without solving a data-dependent optimization problem or gradient computation from scratch. However, its effectiveness is proportional to the amount of data used for computing a universal adversarial perturbation. It requires a large amount of data to achieve a high fooling ratio. In addition,

---

**Algorithm 1: Meta Adversarial Perturbation (MAP)**

**Input:** $\mathbb{D}$, $\alpha$, $\beta$, $f_\theta$, $L$, $\Pi_\epsilon$
**Output:** Meta adversarial perturbations $v$

1  Randomly initialize $v$
2  **while** not done **do**
3      **for** minibatch $\mathbb{B} = \{x^{(i)}, y^{(i)}\} \sim \mathbb{D}$ **do**
4          Evaluate $\nabla_v L(f_\theta)$ using minibatch $\mathbb{B}$ with perturbation $v$
5          Compute adapted perturbations with gradient ascent: $v' = v + \alpha \nabla_v L(f_\theta, \mathbb{B} + v)$
6          Sample a batch of data $\mathbb{B}'$ from $\mathbb{D}$
7          Evaluate $\nabla_v L(f_\theta)$ using minibatch $\mathbb{B}'$ with adapted perturbation $v'$
8          Update $v \leftarrow v + \beta \nabla_v L(f_\theta, \mathbb{B}' + v')$
9          Project $v \leftarrow \Pi_\epsilon(v)$
10     **end**
11 **end**
12 **return** $v$

---

although UAP demonstrates a certain degree of transferability, the fooling ratios on different networks, which are normally lower than 50%, may not be high enough for an attacker. This problem is particularly obvious when the architecture of the target model is very different from the surrogate model (Moosavi-Dezfooli et al. 2017).

Although there are some works (Yang et al. 2021; Yuan et al. 2021) that seem similar to our method, our goal is completely different. (Yuan et al. 2021) proposes to use a meta-learning-like architecture to improve the cross-model transferability of the adversarial examples, while (Yang et al. 2021) devise an approach to learn the optimizer parameterized by a recurrent neural network to generate adversarial attacks. Both works are distinct from the meta adversarial perturbations considered in this paper, as we seek a single perturbation that is able to efficiently adapt to a new data point and fool the classifier with high probability.

## 3 Meta Adversarial Perturbations

We formalize in this section the notion of meta adversarial perturbations (MAPs) and propose an algorithm for computing such perturbations. Our goal is to train a perturbation that can become more effective attacks on new data points within one- or few-step updates. How can we find such a perturbation that can achieve fast adaptation? Inspired by the model-agnostic meta-learning (MAML) (Finn, Abbeel, and Levine 2017), we formulate this problem analogously. Since the perturbation will be updated using a gradient-based iterative method on new data, we will aim to learn a perturbation in such a way that this iterative method can rapidly adapt the perturbation to new data within one or a few iterations.

Formally, we consider a meta adversarial perturbation $v$, which is randomly initialized, and a trained model $f$ parameterized by $\theta$. $L$ denotes a cross-entropy loss and $\mathbb{D}$ denotes the dataset used for generating a MAP. When adapting to a batch of data points $\mathbb{B} = \{x^{(i)}, y^{(i)}\} \sim \mathbb{D}$, the perturbation $v$ becomes $v'$. Our method aims to seek a single meta pertur-

| Attack\Model | | VGG11 | VGG19 | ResNet18 | ResNet50 | DenseNet121 | SENet | MobileNetV2 |
|---|---|---|---|---|---|---|---|---|
| Clean | $\mathbb{D}$ | 100.0% | 100.0% | 100.0% | 100.0% | 100.0% | 100.0% | 100.0% |
| | $\mathbb{T}$ | 92.6% | 93.7% | 95.3% | 95.4% | 95.4% | 95.8% | 94.1% |
| FGSM | $\mathbb{D}$ | 28.0% | 53.0% | 47.0% | 29.0% | 41.0% | 40.0% | 30.0% |
| | $\mathbb{T}$ | 29.3% | 49.4% | 41.4% | 35.7% | 35.5% | 38.2% | 32.8% |
| UAP | $\mathbb{D}$ | 99.0% | 98.0% | 58.0% | 32.0% | 33.0% | 42.0% | 42.0% |
| | $\mathbb{T}$ | 88.9% | 83.3% | 45.8% | 33.5% | 25.5% | 32.5% | 45.8% |
| MAP | $\mathbb{D}$ | 22.0% | 31.0% | 21.0% | 14.0% | 12.0% | 18.0% | 13.0% |
| | $\mathbb{T}$ | 22.0% | 36.1% | 20.3% | 17.4% | 20.8% | 17.6% | 16.3% |

Table 1: The accuracy against different attacks on the set $\mathbb{D}$, and the test set $\mathbb{T}$ (lower means better attacks).

bation $v$ such that after adapting to new data points within a few iterations it can fool the model on almost all data points with high probability. That is, we look for a perturbation $v$ such that

$$f(x + v') \neq f(x) \text{ for "most" } x \sim \mu. \quad (4)$$

We describe such a perturbation *meta* since it can quickly adapt to new data points sampled from the data distribution $\mu$ and cause those data to be misclassified by the model with high probability. Notice that a MAP is image-agnostic, as a single perturbation can adapt to all the new data.

In our method, we use one- or multi-step gradient ascent to compute the updated perturbation $v'$ on new data points. For instance, using one-step gradient ascent to update the perturbation is as follows:

$$v' = v + \alpha \nabla_v L(f_\theta, \mathbb{B} + v), \quad (5)$$

where the step size $\alpha$ is a hyperparameter, which can be seen as $\gamma$ in Eq. (2). For simplicity of notation, we will consider a one-step update for the rest of this section, but it is straightforward to extend our method to multi-step updates.

The meta perturbation is updated by maximizing the loss with respect to $v$ evaluated on a batch of new data points $\mathbb{B}'$ with the addition of the updated perturbation $v'$. More precisely, the meta-objective can be described as:

$$\max_v \sum_{\mathbb{B} \sim \mathbb{D}} L(f_\theta, \mathbb{B}' + v')$$
$$= \max_v \sum_{\mathbb{B} \sim \mathbb{D}} L(f_\theta, \mathbb{B}' + (v + \alpha \nabla_v L(f_\theta, \mathbb{B} + v))). \quad (6)$$

Note that the meta-optimization is performed over the perturbation $v$, whereas the objective is computed using the adapted perturbation $v'$. In effect, our proposed method aims to optimize the meta adversarial perturbation such that after one or a small number of gradient ascent updates on new data points, it will produce maximally effective adversarial perturbations, i.e. attacks with a high success rate.

We use stochastic gradient ascent to optimize the meta-objective:

$$v \leftarrow v + \beta \nabla_v L(f_\theta, \mathbb{B}' + v'), \quad (7)$$

where $\beta$ is the meta step size. Algorithm 1 outlines the key steps of MAP. At line 9, MAP projects the updated perturbations onto the feasible set if they exceed the maximum allowable amount indicated by $\epsilon$. A smaller $\epsilon$ makes an attack less visible to humans.

The meta-gradient update involves a gradient through a gradient. This requires computing Hessian-vector products with an additional backward pass through $v$. Since backpropagating through many inner gradient steps is computation and memory intensive, there are a plethora of works (Li et al. 2017; Nichol, Achiam, and Schulman 2018; Zhou, Wu, and Li 2018; Behl, Baydin, and Torr 2019; Raghu et al. 2019; Rajeswaran et al. 2019; Zintgraf et al. 2019) try to solve this problem after MAML (Finn, Abbeel, and Levine 2017) was proposed. We believe that the computation efficiency of MAP can benefit from those advanced methods.

## 4 Experiments

We conduct experiments to evaluate the performance of MAP using the following default settings.

We assess the MAP on the CIFAR-10 (Krizhevsky, Hinton et al. 2009) test set $\mathbb{T}$, which contains 10,000 images. We follow the experimental protocol proposed by (Moosavi-Dezfooli et al. 2017), where a set $\mathbb{D}$ used to compute the perturbation contains 100 images from the training set, i.e. on average 10 images per class. The maximum allowable perturbation $\epsilon$ is set to $8/255$ measured by $l_\infty$ distance. When computing a MAP, we use one gradient update for Eq. (5) with a fixed step size $\alpha = \epsilon = 8/255$, and use the fast gradient sign method (FGSM) in Eq. (1) as the optimizer. We use seven trained models to measure the effectiveness of MAP, including VGG11, VGG19 (Simonyan and Zisserman 2014), ResNet18, ResNet50 (He et al. 2016), DenseNet121 (Huang et al. 2017), SENet (Hu, Shen, and Sun 2018), and MobileNetV2 (Sandler et al. 2018). We consider FGSM (Goodfellow, Shlens, and Szegedy 2014) and universal adversarial perturbation (UAP) (Moosavi-Dezfooli et al. 2017) as our baselines. We implement baselines using the same hyperparameters when they are applicable.

### 4.1 Non-targeted Attacks

First, we evaluate the performance of different attacks on various models. For the FGSM and MAP, we compute the data-dependent perturbation for each image by using a one-step gradient ascent (see Eq. (1)) to create non-targeted attacks. For the UAP, we follow the original setting as (Moosavi-Dezfooli et al. 2017), where we add the UAP on the test set $\mathbb{T}$ without any adaptation.

The results are shown in Table 1. Each result is reported on the set $\mathbb{D}$, which is used to compute the MAP and UAP,

|  | VGG11 | VGG19 | ResNet18 | ResNet50 | DenseNet121 | SENet | MobileNetV2 |
|---|---|---|---|---|---|---|---|
| VGG11 | **22.0%** | 37.2% | 24.9% | 19.6% | 24.2% | 20.5% | 20.2% |
| VGG19 | 22.9% | 36.1% | 24.5% | 18.3% | 22.0% | 19.2% | 18.3% |
| ResNet18 | 22.7% | 33.6% | **20.3%** | 17.1% | 21.6% | 18.3% | 17.8% |
| ResNet50 | 23.6% | 35.6% | 23.0% | 17.4% | 20.8% | 19.3% | 18.1% |
| DenseNet121 | 23.1% | **32.7%** | 21.3% | **16.1%** | 20.8% | 18.1% | 16.9% |
| SENet | 22.5% | 34.9% | 23.7% | 17.5% | 20.8% | **17.6%** | 17.5% |
| MobileNetV2 | 23.7% | 35.3% | 22.2% | 16.7% | **20.7%** | 18.0% | **16.3%** |
| FGSM | 29.3% | 49.4% | 41.4% | 35.7% | 35.5% | 38.2% | 32.8% |

Table 2: Transferability of the meta adversarial perturbations across different networks (with one-step update on the target model). The percentage indicates the accuracy on the test set $\mathbb{T}$. The row headers indicate the architectures where the meta perturbations are generated (source), and the column headers represent the models where the accuracies are reported (target). The bottom row shows the accuracies of FGSM on the target models without using meta perturbation at initialization.

as well as on the test set $\mathbb{T}$. Note that the test set is not used in the process of the computation of both perturbations. As we can see, MAP significantly outperforms the baselines. For all networks, the MAP achieves roughly 10-20% improvement. These results have an element of surprise, as they show that by merely using a MAP as an initial perturbation for generating adversarial examples, the one-step attack can lead to much lower robustness, compared with the naive FGSM. Moreover, such a perturbation is *image-agnostic*, i.e. a single MAP works well on all test data. We notice that for some models, the UAP performs poorly when only using 100 data for generating the perturbation. These results are consistent with the earlier finding that the UAP requires a large amount of data to achieve a high fooling ratio (Moosavi-Dezfooli et al. 2017).

## 4.2 Transferability in Meta Perturbations

We take a step further to investigate the transferability of MAP. That is, whether the meta perturbations computed from a specific architecture are also effective for another architecture. Table 2 shows a matrix summarizing the transferability of MAP across seven models. For each architecture, we compute a meta perturbation and show the accuracy on all other architectures, with one-step update on the target model. We show the accuracies without using MAP at initialization in the bottom row. As shown in Table 2, the MAP generalizes very well across other models. For instance, the meta perturbation generated from the DenseNet121 achieves comparable performance to those perturbations computed specifically for other models. In practice, when crafting an adversarial example for some other neural networks, using the meta perturbation computed on the DenseNet121 at initialization can lead to a stronger attack, compared with the from-scratch method. The results show that the meta perturbations are therefore not only image-agnostic, but also *model-agnostic*. Such perturbations are generalizable to a wide range of deep neural networks.

## 4.3 Ablation Study

While the above meta perturbations are computed for a set $\mathbb{D}$ containing 100 images from the training set, we now examine the influence of the size $|\mathbb{D}|$ on the effectiveness of

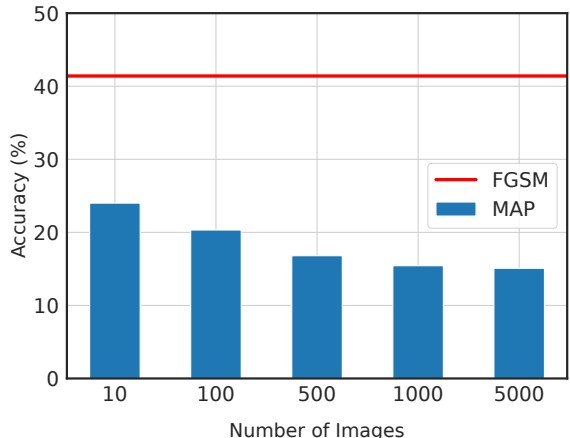

Figure 1: Accuracy on the test set $\mathbb{T}$ versus the number of images in $\mathbb{D}$ for learning MAP.

the MAP. Here we use the ResNet18 for computing the MAP. The results, which are shown in Fig. 1, indicate that a larger size of $\mathbb{D}$ leads to better performance. Surprisingly, even using only 10 images for computing a meta perturbation, such a perturbation still causes the robustness to drop by around 15%, compared with the naive FGSM. This verifies that meta perturbations have a marvelous generalization ability over unseen data points, and can be computed on a very small set of training data.

## 5 Conclusion and Future Work

In this work, we show the existence and realization of a meta adversarial perturbation (MAP), an initial perturbation that can be added to the data for generating more effective adversarial attacks through a one-step gradient ascent. We then propose an algorithm to find such perturbations and conduct extensive experiments to demonstrate their superior performance. For future work, we plan to extend this idea to time-efficient adversarial training (Shafahi et al. 2019; Wong, Rice, and Kolter 2019; Zhang et al. 2019; Zheng et al. 2020). Also, evaluating our attack on robust pre-trained models or different data modalities is another research direction.

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
