# OpenReview forum: "Meta Adversarial Perturbations"
_AAAI.org/2022/Workshop/AdvML — AAAI-22 AdvML Workshop ShortPaper_

### Official Review · Reviewer_sXuN · 2021-11-29
**Meta Adversarial Perturbations**

**Rating:** 6
**Confidence:** 4

**Review:**

This paper proposes a meta adversarial perturbation (MAP) and obtains a better initialization that causes natural images to be misclassified with high probability, which is only updated through a one-step gradient ascent update.

Experimental performance can be demonstrated that the method can better mislead adversarial example classifiers and achieve better performance.

The weakness is also listed as follows:

1.	The authors are encouraged to evaluate the performance on more datasets, such as ImageNet.

2.	More baselines should be introduced for comparison, such as DIM[1] and TIM[2].


[1] Xie et al. Improving transferability of adversarial examples with input diversity

[2] Dong et al. Evading Defenses to Transferable Adversarial Examples by Translation-Invariant Attacks

---

### Decision · Program_Chairs · 2021-12-01

**Decision:**

Accept (Short Paper)

**Comment:**

This paper is accepted as a short paper. Please address the reviewer's comments in the camera-ready version.